# Identification of miR-671-5p and Its Related Pathways as General Mechanisms of Both Form-Deprivation and Lens-Induced Myopia in Mice

**Zedu Cui** [1,2,†]**, Yuke Huang** [1,2,†]**, Xi Chen** [1,2]**, Taiwei Chen** [1,2]**, Xiangtao Hou** [1,2]**, Na Yu** [1,2]**, Yan Li** [1,2]**, Jin Qiu** [1,2]**, Pei Chen** [1,2]**, Keming Yu** [1,2,*] **and Jing Zhuang** [1,2,*]

1    State Key Laboratory of Ophthalmology, Zhongshan Ophthalmic Center, Sun Yat-sen University, Guangzhou 510060, China
2    Guangdong Provincial Key Laboratory of Ophthalmology and Visual Science, Guangzhou 510060, China
*    Correspondence: yukeming@mail.sysu.edu.cn (K.Y.); zhuangj@mail.sysu.edu.cn (J.Z.); Tel.: +86-20-6667-8735 (J.Z.); Fax: +86-20-8733-3271 (J.Z.)
†    These authors contributed equally to this work.

**Abstract:** Animal models have been indispensable in shaping the understanding of myopia mechanisms, with form-deprivation myopia (FDM) and lens-induced myopia (LIM) being the most utilized. Similar pathological outcomes suggest that these two models are under the control of shared mechanisms. miRNAs play an important role in pathological development. Herein, based on two miRNA datasets (GSE131831 and GSE84220), we aimed to reveal the general miRNA changes involved in myopia development. After a comparison of the differentially expressed miRNAs, miR-671-5p was identified as the common downregulated miRNA in the retina. miR-671-5p is highly conserved and related to 40.78% of the target genes of all downregulated miRNAs. Moreover, 584 target genes of miR-671-5p are related to myopia, from which we further identified 8 hub genes. Pathway analysis showed that these hub genes are enriched in visual learning and extra-nuclear estrogen signaling. Furthermore, two of the hub genes are also targeted by atropine, which strongly supports a key role of miR-671-5p in myopic development. Finally, Tead1 was identified as a possible upstream regulator of miR-671-5p in myopia development. Overall, our study identified the general regulatory role of miR-671-5p in myopia as well as its upstream and downstream mechanisms and provided novel treatment targets, which might inspire future studies.

**Keywords:** miRNA; form-deprivation myopia; lens-induced myopia; animal models

## 1. Introduction

Myopia is a kind of refractive disorder with a typical symptom of blurred distance vision, often arising from a mismatch between the axial length and refractive power of the eye. It was reported that myopia is reaching epidemic proportions [1]. Timothy R et al. estimated that approximately 49.8% of the global population will be myopic in 2050 [2]. The global productivity loss associated with the burden of visual impairment resulting from uncorrected myopia was estimated to be USD 244 billion in 2015, with East Asia bearing the greatest potential burden [3]. Moreover, although clear distance vision can be regained with corrections, such as spectacles and refractive surgery, the ocular complications associated with myopia cannot be prevented and can eventually lead to irreversible visual impairment. Thus, the prevention of myopia and its complications have become an important international public health priority [4].

The threat of myopia has prompted a rise in research exploring the causes of this disorder [1]. Experimental myopia has provided a fundamental model for the exploration of myopia mechanisms [5]. Two types of induction, lens-induced myopia (LIM) and form-deprivation myopia (FDM), are commonly performed in various species [6]. LIM is

induced by placing a negative lens in front of the eye, while FDM is established by securing a translucent diffuser over the eye [7]. These two models are different in several aspects, ranging from growth patterns to underlying mechanisms [8]. For instance, LIM exhibits a closed-loop growth pattern, which means that axial elongation stops when compensation is achieved for the given defocus [9], whereas FDM exhibits an open-loop growth pattern [10]. Moreover, FDM tends to be affected by light manipulation, such as constant light and high illuminance [11–13]. Furthermore, the blockage of parasympathetic inputs to the choroid significantly attenuated FDM but not LIM [14]. Despite the dissimilarities, evidence has also shown that some shared mechanisms in the retina–choroid–sclera cascade modulate the two models, inducing similar pathologic changes [8]. Particularly, retinal dysfunction is believed to be the initiating factor in myopia pathogenesis, and some classic myopia pathways shared by both models are located in the retina. For example, research on the dopamine theory has provided the most potent evidence in terms of shared mechanisms. It was proven that retinal dopamine exerts a protective effect against myopic growth via a D2-like mechanism in both models [15]. More importantly, dopamine signaling seemed to play a role in the protective effect of time outdoors, increased illuminance and low-dose atropine [16–19], which suggests that the shared mechanism in the retina of the two models might have general clinical implications. Although dopamine signaling plays an important role in myopia development, it cannot fully explain the onset and progression of myopia, and its upstream and downstream pathways are not fully understood [20]. As such, it is important to compare the two models in terms of other aspects, which could provide novel insights into the general modulatory mechanisms underlying myopia and novel treatment targets with general implications.

The noncoding regulatory system has been shown to play important roles in ocular development and diseases [21]. Among various noncoding RNAs, microRNAs (miRNAs) are the most popular in myopic research. miRNAs are single-stranded noncoding RNAs with approximately 22 nucleotides. They can repress protein expression by degrading target mRNAs or suppressing translation. Because miRNAs act as regulatory hubs in a wide range of biological pathways [22,23], studying the roles of miRNAs in myopia development could provide more information on the general mechanisms. Recently, increasing evidence has shown that miRNAs are involved in both LIM and FDM, with most studies being focused on the roles of retinal miRNAs [6,24–27]. Restricted to species, Tkatchenko et al. performed a microarray analysis on the retinas of both FDM and LIM mouse models and organized nine intersecting miRNA–mRNA pathways mainly involved in neurogenesis regulation [25]. Meanwhile, Tanaka et al. created an miRNA expression profile dataset in LIM mouse models and confirmed the role of Wnt/β-catenin signaling in myopia development [6]. Moreover, they found miR-671-5p downregulation in the LIM mouse models, which was also reported in the FDM mouse models [25]. Thus, these two datasets both reflected various myopia-related pathways in the retina, supporting the reliability of these studies. In addition, some researchers compared the miRNA expression profiles based on these datasets in a meta-analysis study [28], but they did not focus on the shared changes in different paradigms. Tanaka et al. also mentioned the shared miRNA alternation in both paradigms; however, they did not carry out further analysis. As such, we aimed to explore the shared miRNA changes as well as the upstream and downstream regulatory pathways in the retinas of the LIM and FDM mouse models, which might help in understanding the general mechanisms underlying myopia development and shed light on myopia treatment methods.

In this study, we first analyzed the miRNA expression profiles of the FDM and LIM mouse models from two microarray datasets (GSE131831 and GSE84220) and identified miR-671-5p as the common downregulated miRNA in the retinas of both myopia models. The cross-species conservation analysis of its precursor was carried out to confirm its consistency between species. Then, myopia-associated target genes of miR-671-5p were identified and illustrated with a Venn diagram, and a total of 584 genes were recognized. We next performed GO annotation and pathway analysis on these 584 genes to identify

significant biological functions and pathways. Moreover, a protein–protein interaction (PPI) network was established to further understand the interactions between these genes, and the hub genes were identified. Furthermore, we explored the possible associations between miR-671-5p and reliable pharmacological interventions for myopia, namely, atropine. Finally, the latent upstream modulatory mechanisms of miR-671-5p were explored by the in silico prediction of transcription factors in addition to downstream regulatory functions. As such, the in-depth bioinformatics analysis of miR-671-5p revealed the possible mechanisms shared by different myopia models, which might provide insights for future studies.

## 2. Materials and Methods

Data source: Two publicly available miRNA expression profiles were included in our analysis, including one from Tkatchenko's study (GSE84220, the retinas of the FDM mouse model) [25] and another from Tanaka's study (GSE131831, the retinas of the LIM mouse model) [6]. The microarray datasets were acquired from the public Gene Expression Omnibus (GEO) database at https://www.ncbi.nlm.nih.gov/geo/ (accessed on 15 May 2022). The genes were annotated using official gene symbols from the NCBI database and accompanying platform annotation information. The differentially expressed miRNAs from GSE84220 were obtained using the *Limma* R package [29]. We identified 53 miRNAs that were differentially expressed at a significant level ($p$-value < 0.05 and twofold change). The differentially expressed miRNAs from GSE131831 were obtained according to the criteria set by Tanaka et al. (the expression ratio of a fold-change of >2 and <0.5 with a detected flag in both the denominator and the numerator or a fold-change of >5 and <0.2 with no detected flag in either the denominator or the numerator). We hence identified nine miRNAs. The analyzed data we used in this research were all taken from the public source; hence, the Animal Ethics Committees' approval was not required.

Prediction of candidate miRNA targets: As miRNAs function by downregulating the expression of target genes, predicting miRNA targets using bioinformatics is crucial for understanding miRNA function. Using the online tools of miRWalk [30], several machine learning algorithms predicted the target genes of differentially expressed miRNAs. Potential target genes were chosen based on whether or not they satisfied the following criteria [31]: (1) they had miRNA binding sites with a binding probability >0.99, and (2) the predicted miRNA/target binding energy was less than −20 kcal/mol.

Conservation analysis of miRNA. Mature sequences from 12 animal species of miR-671-5p were obtained in the miRBase database (version 22.1, http://microrna.sanger.ac.uk/sequences/) (accessed on 21 July 2022) [32]. Then the base comparisons of the seed region and the full sequence were performed manually.

Drug–gene interaction analysis. We investigated genes that are associated with atropine based on the Comparative Toxicogenomics Database (CTD (http://ctdbase.org/)) (accessed on 21 July 2022). Chemical–gene, chemical–disease, and gene–disease interactions from the literature are available in the CTD [33]. The negative regulation between the drug and the gene was filtered out because atropine should have a therapeutic effect and inhibit dysregulated myopic genes.

Collection of disease-related targets: Myopia-related targets were searched for through two disease-related databases: DisGeNET [34] and GeneCards. Then, they were merged after deleting the repetitive ones.

PPI network construction: STRING [35] is commonly used for PPI prediction. The PPI network was constructed using the protein interaction scores that were filtered out and below 0.9 (high confidence). Cytoscape was used to enhance the PPI network's appearance [36]. The topological network's properties were examined using Cytoscape's NetWorkAnalyzer tools [37]. NetWorkAnalyzer is mainly used to analyze network parameters. The fit of the distribution of the degree was monitored using the poweRlaw package in R [38]. The score of the co-efficient of determination was then calculated. A Cytoscape plugin called cytoHubba [39] thoroughly builds sub-networks and detects the hub elements of the PPI network. We used three local-based algorithms (Maximal Clique Centrality,

Density of Maximum Neighborhood Component, Degree Method) and three global-based algorithms (BottleNeck, Edge Percolated Component, Closeness) to increase accuracy because it makes sense to employ multiple methods for detecting essential proteins.

Gene annotation and pathway enrichment analysis: To analyze high-throughput molecular data regarding the underlying biological processes, Gene Ontology (GO) enrichment analysis, including BP (Biological Process), MF (Molecular Function) and CC (Cellular Component), is utilized. The GO enrichment analysis of eight hub genes was carried out using the clusterProfiler package [40] in the R software. The pathway information of eight hub genes was obtained from three pathway databases: WikiPathways, KEGG and Reactome, and it was carried out using the clusterProfiler package. Pathway enrichment techniques statistically check the biological pathways for over-representation in the gene list relative to what is predicted by chance. The obtained GO terms and pathways with *p*-values < 0.05 were considered significantly enriched. The hierarchical tree was further examined using the Cytoscape plugin ClueGO, which could classify the non-redundant GO terms [41]. The GO tree interval was set from three to eight, and the kappa score was set to 0.4.

Transcriptional factor–miRNA co-regulatory analysis: Transcription factors (TFs) that are likely to be related to miR-671-5p were predicted using two methods: (1) directly predicted via the TransmiR v2.0 database [42] and (2) indirectly predicted based on Chpf2, the host gene of miR-671-5p [43], via the Pscan database [44].

Collection of putative transcription factor-targeted genes: Cistrome DB (http://cistrome.org/db/#/) (accessed on 25 July 2022) [45] compiles data from ChIP-seq and chromatin accessibility measurements. All samples were acquired from the Roadmap Epigenomics, GEO and ENCODE databases, and data visualization functions and extensions were involved in supporting the putative target genes of TFs. The genes associated with Tead1 were those with scores greater than 1.

## 3. Results

### 3.1. The Analytical Pipeline Reveals the General Regulatory Role of Retinal miR-671-5p in Both LIM and FDM Mouse Models

To investigate whether the pathological processes induced by LIM shared similarities with FDM at the posttranscriptional level, we obtained differentially expressed miRNAs in the mouse retina from two raw microarray datasets (GSE131831 and GSE84220). By calculating the expression ratio, upregulated miRNAs (0 in LIM and 37 in FDM) and downregulated miRNAs (9 in LIM and 16 in FDM) were identified (Figure 1A). Importantly, miR-671-5p expression was concurrently downregulated in both LIM and FDM models (Figure 1B). This result indicates that miR-671-5p might mediate some general pathways in both paradigms.

miRNAs predominantly affect multiple downstream target genes and therefore exert posttranscriptional regulatory functions [22]. Therefore, the key to understanding the mechanism of action of miRNAs is to understand the interactions between miRNAs and their target genes. Accordingly, there were 7013 overlapping mRNAs targeted by downregulated miRNAs in both the LIM and FDM mouse models. Among them, miR-671-5p targeted 2860 mRNAs, accounting for 40.78% of the overlapping mRNAs (Figure 1C). To further evaluate the extensiveness of miR-671-5p regulation, the cross-species conservation analysis of its precursor was accessed via BLAST analysis in the NCBI database. We found that the seed region of miR-671-5p preserved a high consistency in up to 12 species, including *Mus musculus*, *Homo sapiens*, *Macaca mulatta*, *Pongo pygmaeus*, *Equus caballus*, *Oryctolagus cuniculus*, *Cavia porcellus*, *Sus scrofa*, *Cricetulus griseus*, *Capra hircus*, *Tupaia chinensis* and *Dasypus novemcinctus* (Figure 1D). Notably, the sequences of *Homo sapiens* were strictly identical with those of *Mus musculus*. Since highly conserved miRNAs often exert similar functions in various species, these results indicate that miR-671-5p is functionally conserved among species. Therefore, miR-671-5p may play a key role in the development of myopia due to its extensive regulatory effects and high conservation.

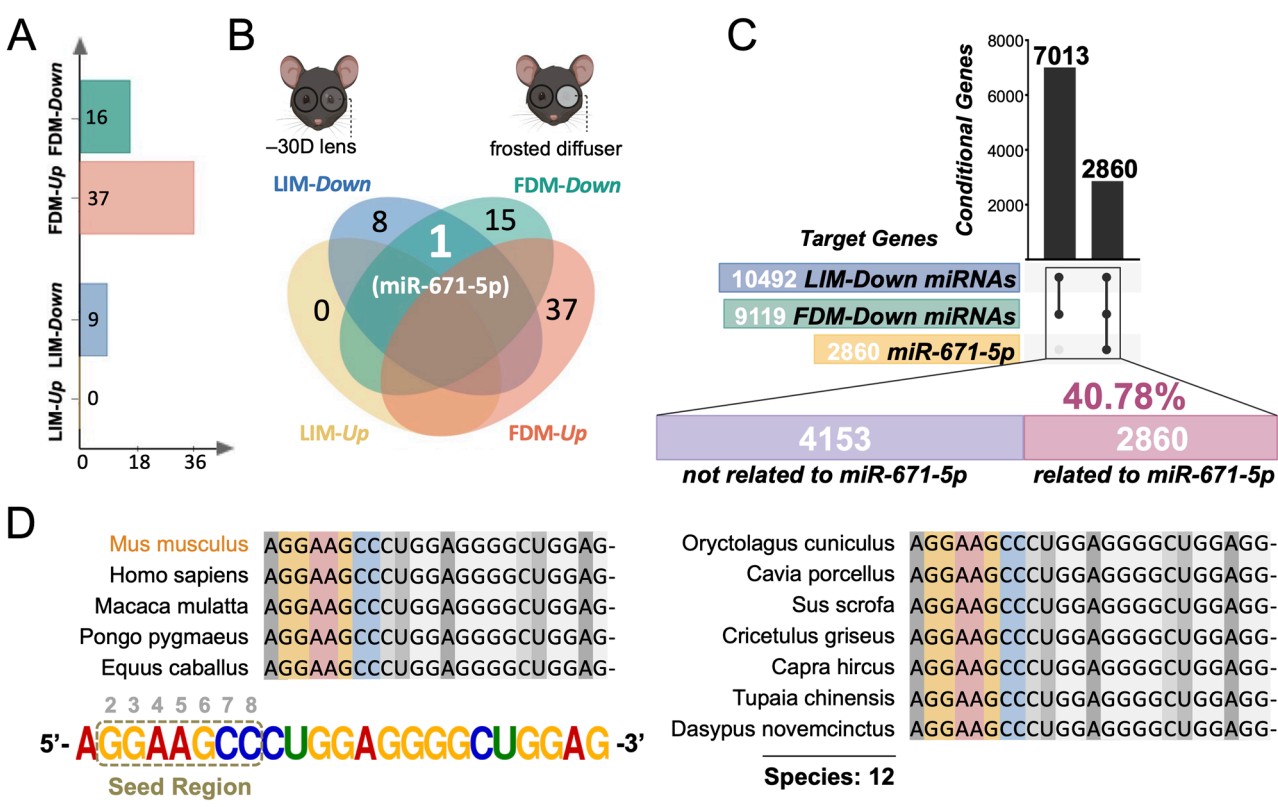

**Figure 1.** Identification and characterization of miR-671-5p as the shared downregulated miRNA in LIM and FDM mouse models. (**A**) Differentially expressed miRNAs in the retinas of LIM and FDM mouse models. The numbers on the bars represent the total differentially expressed miRNAs in each group. X-axis: numbers of differentially expressed miRNAs. Y-axis: the name of each group. (**B**) miR-671-5p is concurrently downregulated in the retinas of both LIM and FDM mice. (**C**) A total of 40.78% of all the target genes of the downregulated miRNAs in the LIM and FDM retinas are related to miR-671-5p. (**D**) Comparison of miR-671-5p seed sequences between 12 different species.

*3.2. Identifying Hub Myopia-Related Genes Regulated by miR-671-5p*

To elucidate the contribution of miR-671-5p to myopia pathogenesis, we utilized a Venn diagram and found 584 overlapping genes between miR-671-5p target genes and myopia-associated genes (Figure 2A). Based on the 584 identified target genes, we constructed a PPI network consisting of 214 protein nodes and 428 PPI edges (Supplementary Figure S1). We calculated the fit of the distribution of the number of edges per node, which is called its degree, to the power law distribution. The degree followed a long-tailed distribution ($R^2$ = 0.940) (Figure 2B), indicating that only a few proteins (hub proteins) communicated with most other nodes in the PPI network, while the majority of proteins had only a few connections. These few proteins were hence considered to play major roles in controlling miR-671-5p functional networks. Therefore, we further explored the hub genes using the cytoHubba plug-in by jointly analyzing the local and global features of the network (Figure 2C). Maximal Clique Centrality, Density of Maximum Neighborhood Component and Degree were the local-based parameters. Included among the global-based parameters were BottleNeck, Edge Percolated Component and Closeness. The intersection of the highest-ranked genes was histone deacetylase 1 (Hdac1), mitogen-activated protein kinase 1 (Mapk1), methyl-CpG binding protein (Mecp2), actin-like 6a (Actl6a), huntingtin (Htt), G protein subunit beta 5 (Gnb5), Runx family transcription factor 2 (Runx2) and cAMP responsive element binding protein 1 (Creb1) (Figure 2C). Thus, these eight hub genes may play a dominant role in miR-671-5p-mediated retinal dysfunction in myopia pathogenesis.

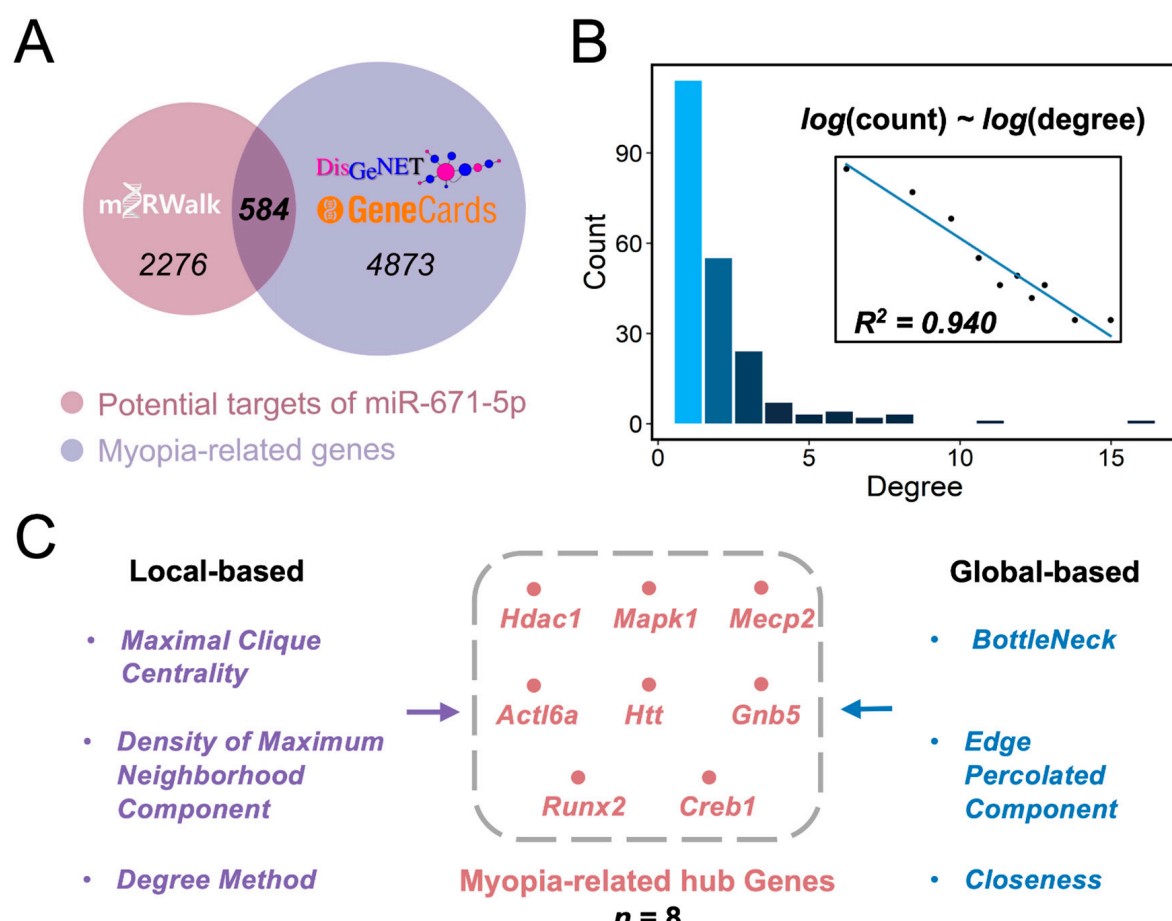

**Figure 2.** Identifying hub myopia-related target genes of miR-671-5p. (**A**) Venn diagram shows that a total of 584 miR-671-5p target genes are associated with myopia; (**B**) Node degree distribution shows that the PPI network follows a typical long-tail distribution; (**C**) The top eight hub genes ranked according to the calculation of node connectivity by CytoHubba.

*3.3. Integrated Analysis Reveals the Biological Functions of the Hub Genes*

We next performed GO annotation and pathway enrichment analysis to identify the potential mechanisms of the eight hub genes in myopia pathogenesis. The results of the annotation indicated that these genes participated in several cell processes, including "Gliogenesis", "SWI/SNF superfamily-type complex" and "p53 binding" (Figure 3A). Additionally, pathway analysis revealed that these genes were enriched in "ESR-mediated", "Extra-nuclear estrogen", "Nuclear receptors", "Kinase and transcription factor activation" and "MAPK targets/Nuclear events mediated by MAP kinases" signaling pathways (Figure 3B). To comprehensively explore the interrelations of functionally grouped terms, we further reduced redundancies by removing terms that were subsets of other terms by a simple clustering algorithm relying on semantic similarity measures. The GO annotation term converged onto "Visual learning" (Figure 3C). Highly ranked gene markers included Htt, Creb1 and Mecp2. The pathway enrichment term converged onto "Extra-nuclear estrogen" signaling (Figure 3C). Highly ranked gene markers included Gnb5, Mapk1 and Creb1. Taken together, strong associations between these hub genes and visual function were identified, indicating that these genes might play vital roles in myopia development.

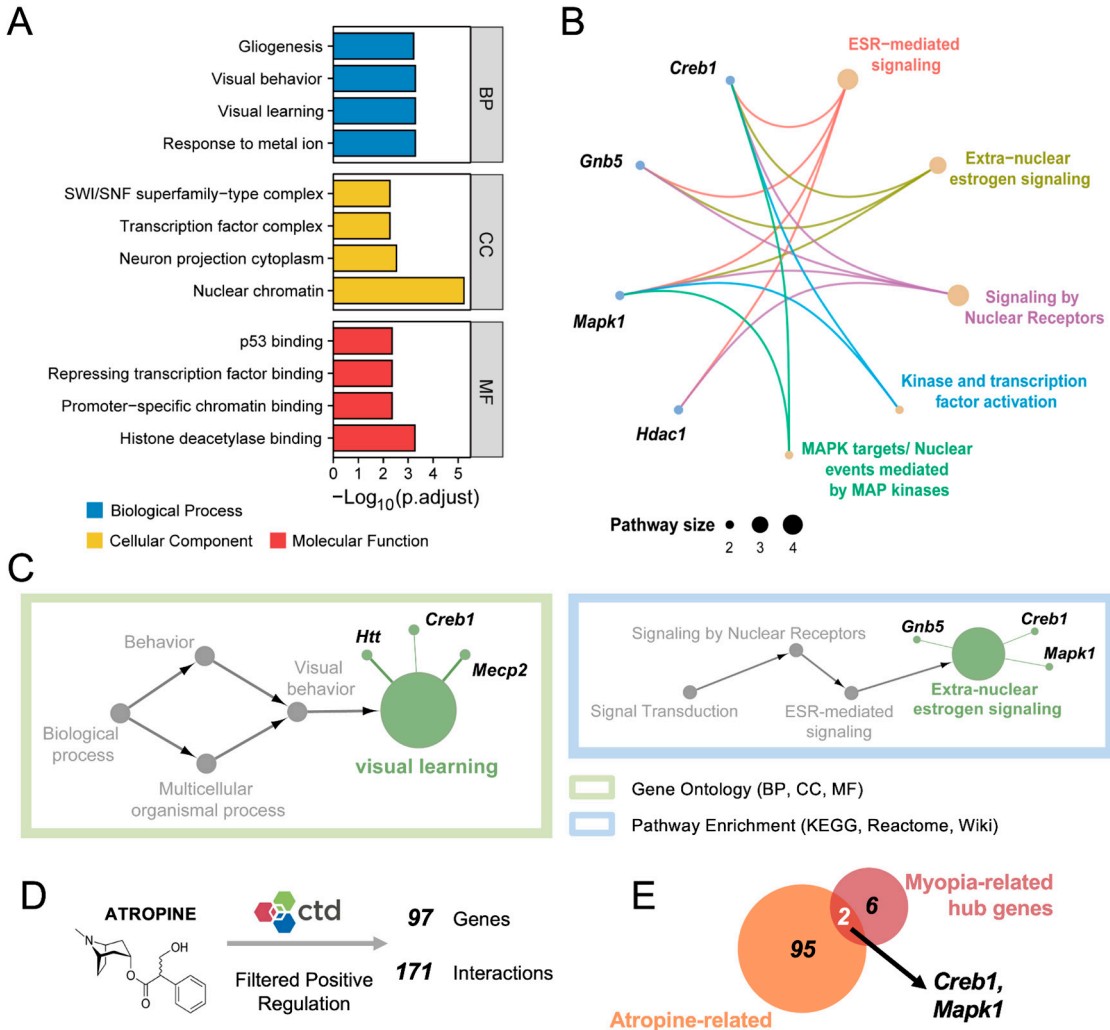

**Figure 3.** GO annotation and pathway enrichment analysis based on the top hub genes. (**A**) Selected enriched GO terms of the top eight hub genes. The colored bars show the top four ranked GO terms in BP, CC and MF, respectively; (**B**) Pathway enrichment analysis of the eight hub genes. The blue nodes represent genes, and the yellow nodes represent pathway names. The larger the nodes' size, the smaller the value of *P*. The branches indicate the relationship between genes and pathways. Branches of the same color come from the same pathway and point to their different follower genes; (**C**) GO terms and enriched pathways are summarized based on semantic similarity measures; (**D**) Atropine-targeted genes from the CTD database: 97 genes could be downregulated by atropine treatment; (**E**) Venn diagram shows that two out of the eight hub genes are negatively related to atropine treatment. BP: Biological Process, MF: Molecular Function, CC: Cellular Component, CTD: Comparative Toxicogenomics Database.

Next, we further explore the biofunctions of vital hub genes in myopia development. Low doses of atropine are known to slow the progression of myopia, which is the oldest and most effective pharmacological treatment in myopia control [46]. Although atropine can partially retard myopia progression via the anti-accommodative mechanism [47], some evidence has suggested that atropine treatment also can modulate several myopia-related pathways in retina and sclera. Additionally, Wang et al. find that the levels of atropine in the retina begin to increase three hours after topical application and for at least 24 h [48]. Thus, we identified the associations between the hub genes and atropine treatment. There were 97 identified genes from the CTD database that could be downregulated by atropine (Figure 3D). Among these genes, Mapk1 and Creb1 were also miR-671-5p target genes

(Figure 3E). These results gave a preliminary supposition that Mapk1 and Creb1 might play a key role in myopia pathogenesis from a distinctive perspective.

### 3.4. Tead1 May Act as an Upstream Regulator of miR-671-5p in Myopia Pathogenesis

Transcription factor (TF)–miRNA regulation is the key effector of eukaryotic gene control. To determine the upstream regulator of miR-671-5p, we predicted several TFs based on the precursor sequences and one of the identified host genes, chondroitin polymerizing factor 2 (Chpf2) (Figure 4A). Combined with myopia-related TFs, Tead1 was the only TF intersected by the three groups (Figure 4B). The sequence of the Tead1 binding site was ACATTCCAG (Figure 4C). Although the inhibitory effects of miR-671-5p on its target genes have been well established, the regulatory patterns between miR-671-5p and Tead1 are still underexplored (Figure 4D). Moreover, we found that Tead1 could also directly bind to the target genes Hdac1, Mapk1 and Gnb5. Taken together, Tead1 might be the upstream regulator of miR-671-5p and several target genes in myopia development. However, the clear and specific role of the miR-671-5p regulatory network in myopia development has not yet been completely elucidated, and in-depth studies in this field are highly warranted.

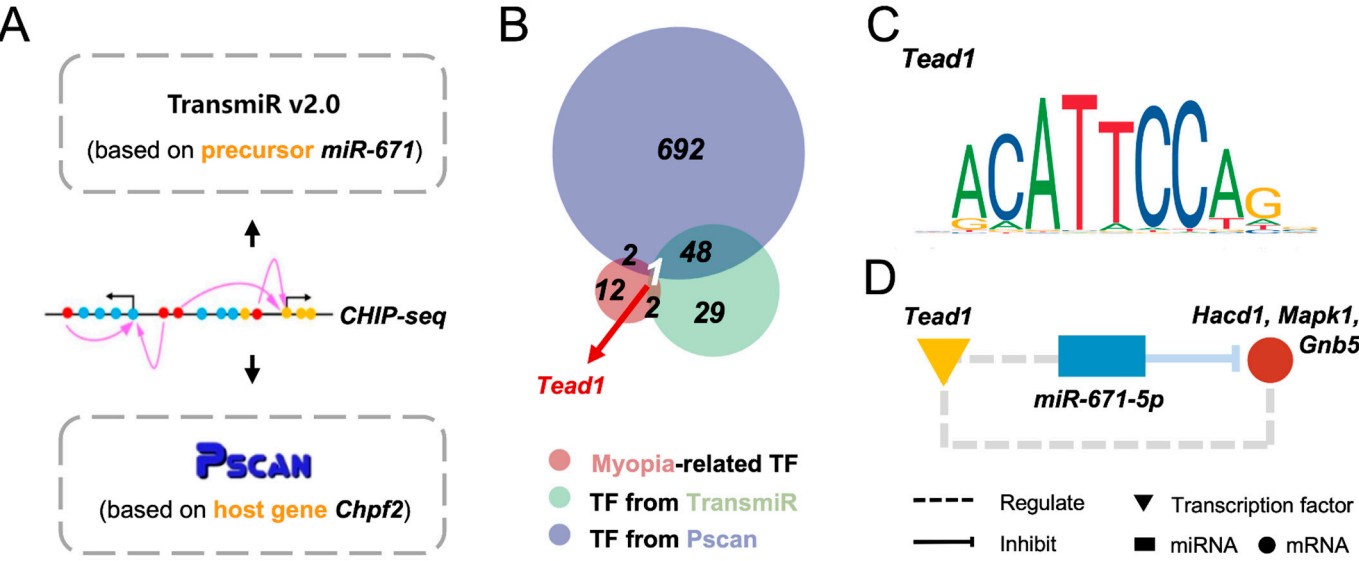

**Figure 4.** Prediction of the upstream regulators of miR-671-5p in myopia pathology. (**A**) Prediction of the upstream TFs of miR-671-5p based on its precursor and host gene; (**B**) Intersection of myopia-related and predicted miR-671-5p upstream TFs: Tead1 might act as the upstream regulator of miR-671-5p in myopia pathology; (**C**) The sequence of the binding site of Tead1 and miR-671-5p; (**D**) Potential regulatory patterns between Tead1, miR-671-5p and the hub target genes.

## 4. Discussion

The exploration of similar noncoding pathway changes between the two major forms of experimental myopia might provide novel insights into general myopia pathogenesis. In this study, we found that the FDM and LIM mouse models are similarly influenced by the downregulation of retinal miR-671-5p, indicating its vital role in myopia development. Downstream myopia-related targets of miR-671-5p were identified, with eight hub genes that might dominate the regulatory network. Importantly, among these hub genes, Mapk1 and Creb1 are downstream effectors of atropine treatment, which further confirms that miR-671-5p plays a dominant role in myopia pathogenesis. In addition to the downstream mechanisms, Tead1 was identified as a possible upstream regulator of miR-671-5p in myopia development. Thus, our study suggests the general regulatory role of retinal miR-671-5p in myopia and deduces its potential upstream and downstream mechanisms by in-depth bioinformatics analysis, which might provide novel insights for myopia pathogenesis and clinical treatments.

In our study, we found that miR-671-5p was downregulated in both the FDM and LIM mouse models. miR-671-5p has been recently reported as a tumor suppressor. It is negatively correlated with the levels of TGF-β and HIF-1α, which are prevalent myopic biomarkers [49]. Additionally, it suppresses the PI3K/AKT signaling pathway, therefore inhibiting tumor progression in the digestive system [50]. Although these proteins and pathways are closely associated with myopia, there have been no studies on the effect of miR-671-5p on myopia. Importantly, our results showed that, among all miRNA-targeted genes obtained from the two myopic mouse models, a large percentage (40.78%) were targets of miR-671-5p (Figure 1C). This result indicated that miR-671-5p might be implicated in multiple biological pathways associated with myopia development. Moreover, the seed region of miR-671-5p shared an identical sequence among 12 species, including humans (Figure 1D). Highly conserved miRNAs often have similar functions in different animal lineages [51]. Collectively, one strength of our study is the identification of miR-671-5p as a general biomarker of myopia in both paradigms.

By using GO terms and pathway databases, we interpreted the underlying mechanisms of miR-671-5p-targeted hub genes. Gliogenesis is one of the enriched GO entries (Figure 3A), which is indirectly supported by previous studies. It has been reported that the FDM retina tends to have increased numbers of astrocytes and Müller cells [52]. The secreted serine protease PRSS56 derived from Müller cells was shown to contribute to excessive ocular axial elongation [53]. Following hyperproliferation, the retina becomes hypoxic and may further trigger the scleral hypoxic cascade reaction [54]. Moreover, coupled with semantic similarity measures, our results showed that extra-nuclear estrogen signaling ranked highest in pathway enrichment (Figure 3C). In line with our findings, current clinical research has shown that higher levels of estrogen are more likely to cause refractive errors [55,56]. The changes in sex hormone levels can alter visual sensitivity [57] and lower light-adaptation processes for some females [58]. Estrogen receptors (ERs) have been demonstrated in the human retina, suggesting a possible direct effect of estrogen on retinal pathological changes [59]. Therefore, it is proposed that estrogen-involved signaling may trigger retinal dysfunction, ultimately causing myopia.

Our results also indicated important roles of Mapk1 and Creb1 in myopia development. These two hub genes were associated with extra-nuclear estrogen signaling (Figure 3C). Meanwhile, we found that they are potential target genes of atropine, the classic and most recommended pharmacological treatment against myopia [47] (Figure 3E). The ligand-independent activation of ERs has been implicated in MAPK cascades. Under epidermal growth factor (EGF) stimulation, MAPK can activate ERα by directly phosphorylating Ser118 [60,60]. Moreover, considering that ERs can be activated in the absence of the ligand estrogen, it has been reported that cAMP-Creb1 stimulates and is required for ERα activity [61,62]. In addition to their associations with estrogen signaling, Mapk1 and Creb1 are involved in several other myopia-related bioactivities in the retina. First, Mapk1 is actively involved in retinal development [63]. It was reported that Hif-1α could activate MAPK signaling and thereby stimulate scleral collagen remodeling in pathologic myopia [64]. Similarly, Creb1 is a pleiotropic leucine zipper transcription factor that is essential for neurogenesis and retinal development [65]. It can be activated by intracellular cAMP in response to neurotransmitters, such as dopamine [66]. Although cAMP and dopamine exert potent bidirectional effects on myopic eye growth [67], it is likely that Creb1 may not act as their downstream effector in myopia development, since it is supposedly upregulated in myopic retinas. Accordingly, Mapk1 and Creb1 might be the important signals in the retina–choroid–sclera cascade. Nevertheless, the association between retinal Mapk1 and Creb1, myopia and atropine treatment needs further experimental validation, and future studies are required to explore the exact effects. Taken together, these bioinformatic results provide directions for future studies on the impact of miR-671-5p and its target genes on myopia.

In addition to the downstream mechanisms of miR-671-5p, we identified Tead1 as an upstream regulator of miR-671-5p in terms of myopia modulation by in silico predic-

tion (Figure 4). Tead1 is a transcription factor associated with multiple myopia-related bioactivities and pathways, such as glycolysis, Wnt signaling and cellular growth [68–71]. Specifically, an altered glycolysis status can induce the proliferative activity of YAP/TAZ by interacting with Tead1 in tumors [68]. Similarly, Tead1 can enhance neuronal growth and thereby play an important role in the development of neural systems [72]. Although most of these functions were validated in non-ocular tissues, the mutation of Tead1 was responsible for pathologic syndromes, with myopia as a manifestation [73–75], which indicates that Tead1 might play a role in the local regulatory networks. As such, it is possible that Tead1 signaling is influenced by abnormal glycolysis metabolism in the myopic retina, thereby regulating miR-671-5p signaling and downstream bioactivities, such as neurite outgrowth. However, Tead1 is the calculated and predicted TF of miR671-5p, and it is not verified by experimental evidence currently. Further experiments such as chromatin immunoprecipitation (ChIP) and luciferase report assays are required to explore the exact regulatory relationship between Tead1 and miR-671-5p.

## 5. Conclusions

In conclusion, our study demonstrated a general regulatory role of miR-671-5p in various myopia models, along with its upstream and downstream mechanisms. Moreover, the overlapping downstream effectors of miR-671-5p and atropine treatment were identified. These findings have provided a new molecular theoretical basis of myopia pathogenesis and atropine mechanisms, which might inspire future studies and help develop novel myopia treatments.

**Supplementary Materials:** The following supporting information can be downloaded at: https://www.mdpi.com/article/10.3390/cimb45030132/s1, Figure S1: The PPI network of the identified target genes.

**Author Contributions:** Conceptualization, Z.C.; Data curation, T.C.; Formal analysis, N.Y. and Y.L.; Funding acquisition, P.C. and K.Y.; Investigation, X.H.; Methodology, X.C.; Software, X.C.; Supervision, K.Y. and J.Z.; Validation, J.Q.; Visualization, Y.H.; Writing—original draft, Z.C. and Y.H.; Writing—review & editing, J.Z. All authors have read and agreed to the published version of the manuscript.

**Funding:** This work was funded by Guangzhou Municipal Science and Technology Project (Grant No. 201803010091 (K.Y.)), and National Natural Science Foundation of China (Grant No. 81900850 (P.C.)).

**Institutional Review Board Statement:** Not applicable.

**Informed Consent Statement:** Not applicable.

**Data Availability Statement:** The data that support the findings of this study are available from the corresponding author upon reasonable request. The datasets analyzed during the current study are available in the NCBI repository (https://www.ncbi.nlm.nih.gov/geo/) (accessed on 15 May 2022).

**Conflicts of Interest:** The authors declare no conflict of interest.

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
