# Peer review of "Identification of miR-671-5p and Its Related Pathways as General Mechanisms of Both Form-Deprivation and Lens-Induced Myopia in Mice"

_cimb, doi:10.3390/cimb45030132_

Round 1

Reviewer 1 Report

Review for: “miR-671-5p-related Molecules are Potential Retinal Biomarkers in Various Models of Experimental Myopia”

In the present study, the authors perform an in silico analysis of mIR-671-5p in myopia and used different bioinformatic tools  to give insight on potential downstream and upstream targets, of this mIR. The message of this manuscript is clear with excellent and easy to understand figures that help convey this study. Even so, this manuscript present some major flaws that need to be addressed before being published.

Major comments:

Title. The core text and conclusions do not match the manuscript title. Title must be completely rewritten to match this article and should not include “molecules”, “potential retinal biomarkers” or “various models of experimental myopia”.

Lack of evidenced problem and hypothesis on this manuscript. Please ensure that this is addressed in the introduction of the manuscript.

miR-671-5p downregulation in LIM and FDM models. While the initial results suggest that the authors investigated and found miR-671-5p as the commonly downregulated mIR, this findings are not novel. The authors of the LIM microarray (doi: 10.3390/ijms20153629) had previously reported these findings and should therefore be acknowledge for that. Please amend the document accordingly.

Reference list are not formatted according to the manuscript guidelines.

Supplementary figure is difficult to interpret and legend should be more descriptive.

Minor comments:

Line 43. The authors have yet to say what LIM and FDM stands for.

Line 190. A) Authors should mention it is the number of differentially expressed miR and also include that information in X axis of Figure A

Line 283. “Thus, our study identified the general regulatory role of retinal miR--671-5p in myopia as well as its upstream and downstream mechanisms”. The authors need to take into account that this is an in silico study and use more suggestive language and that this findings are mostly hypothesis at this point.

Author Response

Please see the attachment. The response to Reviewer 1 is uploaded as Word file.

Reviewer 2 Report

It was a pleasure reviewing this very interesting and informative study. I am not a biologist, so my understanding of the subtleties of the biologic methods, etc, is limited. However the paper is well written and clearly presented, and difficulties appear to have been anticipated and addressed. 

There can be no doubt that acquired myopia represents retinal disorder induced by functional hyperopia that is chemically mediated and leads to lengthening of the globe to achieve focus. The authors give insights into the biochemical processes and key molecules and possible role of mRNA is this process.

My only criticism, if it is one, is connecting the efficacy of juvenile atropine to various candidate retinal receptors or gene responders to atropine. Atropine works fundamentally (and almost certainly sufficiently) by preventing accommodation. As the authors point out in other contexts, the fact that there are other potential biologic responders to atropine does not mean that they have any role in the therapeutic effectiveness of atropine for preventing myopia. Association, no matter how attractive, is not causation and these are the kind of connections that can lead us far astray.  Please add a brief disclaimer here as to the relevance of these atropine sensitive elements.

Author Response

Please see the attachment. The response to Reviewer 2 is uploaded as Word file.

Reviewer 3 Report

This manuscript describes the miR-671-5p-related molecules as potential retinal biomarkers in myopia mouse model.

The study design is clear; however, additional experiments need to be completed to better support the hypothesis in this manuscript. Here are detailed comments:

1.       How did the differentially expressed miRNAs were identified? Please describe in detail in the methods.

2.       0 upregulated miRNA in LIM was identified, Figure1B is confusing.

3.       The authors should explain in detail how the Figure 3B was generated. What does branch and length of branch mean. How was the GO enrichment analysis done?

4.       What dose CTD stands for?

5.       Besides the computational prediction, is there any other evidence supporting Tead1 is the bona fide TF for miR-671-5p?

6.       Overall, the methods part needs more details so other reader can repeat the analysis if necessary.

7.       The references needs to be in the same style.

Author Response

Please see the attachment. The response to Reviewer 3 is uploaded as Word file.

Round 2

Reviewer 1 Report

Thank you for addressing my comments.

No major issues remain.

Reviewer 3 Report

The authors have addressed most of my concerns. I think it can be accepted.